# Comparison of perceived stress and oral health status using perceived stress scale and DMFT index between healthy, undiagnosed hypertensive, and known hypertensive dental patients

Saira Atif [1,2]*, Uffaq Syed [1,2], Musarat Rafiq[1,2], Ateeb Fatima[1,2], Sadia Rana[3], Madiha Tariq [1,2]

1 Combined Military Hospital Lahore Medical College & Institute of Dentistry, Lahore, Pakistan, 2 National University of Medical Sciences, Rawalpindi, Pakistan, 3 Sharif Medical & Dental College, Lahore, Pakistan

* sairaamin80@gmail.com

**Data Availability Statement:** A de-identified dataset has been supplied in the Supporting

## Abstract

Subtle and asymptomatic nature of high blood pressure results in increase in mortality and morbidity. Dentists may play a vital role in identifying patients with suspected hypertension who are not yet diagnosed to refer them timely to physicians for diagnosis and management of the condition. The aim of this study was to compare the oral health status using decayed, missing, filled teeth (DMFT) index and perceived stress score using perceived stress scale (PSS-10) between systemically healthy dental patients, and patients with undiagnosed and known hypertension attending dental out-patient department of Combined Military Hospital (CMH) Lahore Medical College & Institute of Dentistry, Lahore, Pakistan. It is a cross-sectional descriptive study in which 108 patients participated selected through purposive sampling method, 36 in each group. According to Kruskal Wallis H test, there were significant associations seen among the three groups with age (p < 0.001), DMFT (p < 0.001), and PSS-10 scores (p = 0.003). According to Spearman's matrix correlations, moderate positive correlations were observed between mean arterial pressure (MAP) and DMFT (r = 0.46, p < 0.001), and MAP and age (r = 0.38, p = 0.001), DMFT and PSS-10 (r = 0.47, p < 0.001), and DMFT and age (r = 0.33, p = 0.004) in healthy and patients with undiagnosed hypertension. It may be concluded that patients with raised blood pressure have higher perceived stress and deteriorated oral health.

## Introduction

Hypertension is a global health concern [1]. High blood pressure is one of the major challenges that Pakistan healthcare system is facing [2, 3]. Globally, it has been reported that 41% of women and 51% of men with hypertension were not previously diagnosed with the condition [1]. Early detection of high blood pressure can help reduce cardiovascular complications [4]. Most of the people are unaware of hypertension because of absence of any warning signs or symptoms [5].

Information files. The patient data file has potential sensitive information with a possibility of identification of study participants due to small sample size and study duration mentioned in the manuscript. Ethical Review Committee restricts public sharing of data of Combined Military Hospital due to privacy issues. Interested researchers may contact Chairperson Ethical Review Committee of CMH Lahore Medical College & Institute of Dentistry at erc@cmhlahore.edu.pk for obtaining permission to have access to data.

**Funding:** The author(s) received no specific funding for this work.

**Competing interests:** The authors have declared that no competing interests exist.

Blood pressure is now considered a vital sign which is measured in all standard healthcare settings. However, in a developing country like Pakistan, regular healthcare visits for screening or check-ups are rare, and people visit such facilities only in case of a health problem [6]. Dentists are integral part of oral healthcare system and their role is not only performing quality dental care, but also of screening and referring patients for any suspected systemic disease such as oral cancers, diabetes, and hypertension which could lead to morbidity or mortality if left untreated. During dental visits, only 13.3% of dentists in Saudia Arabia measured blood pressure of all patients before treatment [7]. Studies have reported that dentists play an important role in referring patients with suspected hypertension to primary health care facility, out of which a considerable number are subsequently diagnosed with hypertension [8, 9]. In a study conducted in a diagnostic center, faculty of dentistry in Cairo, Egypt, it was reported that 14.9% of the patients screened were not previously diagnosed with hypertension [10]. Individuals with high blood pressure are at risk of developing medical complications during or after dental treatment and their health status may affect the dental treatment outcomes [11].

One of the risk factor of hypertension in Pakistani population is stress and anxiety [2] which may cause alteration in blood pressure levels [12]. Chronic stress may play a role in the development of hypertension and cardiovascular diseases [13]. It has been reported that prolonged and chronic stress can have an effect on the development of hypertension [14]. Studies have reported that in patients with hypertension, those with multiple dental caries were at risk of cardiovascular complications [15, 16]. Moreover, in a longitudinal research, positive association was reported between the number of lost teeth and the risk of hypertension [17]. It is possible that hypertension and dental health are associated; it may either create or exacerbate oral health issues [18, 19]. There is limited data available on hypertension, perceived stress and oral health status of a subset of Pakistani population.

The aim of this study was to compare the perceived stress scores using a validated tool known as perceived stress scale (PSS-10) and oral health status using decayed, missing, filled teeth (DMFT) index between patients with undiagnosed hypertension, known hypertension, and healthy controls.

## Materials and methods

### Study design and setting

This cross-sectional descriptive study was conducted in dental out-patient department of CMH Lahore Medical College & Institute of Dentistry, Lahore, Pakistan from November 1 to December 31, 2023.

### Ethics statement

Study was approved by Ethical Review Committee of CMH Lahore Medical College & Institute of Dentistry (605/ERC/CMH/LMC) and adhered to principles of Declaration of Helsinki. Participants were recruited from the dental out-patient department of Institute of Dentistry. The purpose of the study was explained to the individuals, they were provided time to discuss any queries with one of the investigators. Participants were assured that the data will be kept confidential, and they have the option to withdraw from the study at any stage. Written informed consent was obtained from all participants.

### Study participants

Study participants were recruited from dental out-patient department using purposive sampling method and were placed into either of the following three groups: healthy, undiagnosed

hypertensive, or known hypertensive. Relevant medical and family history and initial blood pressure measurement was taken to screen potential participants for eligibility based on inclusion criteria stated ahead. Individuals who met the criteria and were willing to participate were selected. Sample size was calculated based on previous study considering 95% confidence level and power of study as 90%, using mean difference reported in DMFT in hypertensives (9.9 ±8.4) and healthy controls (3±3.6) [15], the calculated sample size was 18 in each group. However, sample size was increased to 36 in each group to increase robustness in statistical analysis. Data were recorded from all participants meeting the inclusion and exclusion criteria. Inclusion criteria were patients above 21 years of age for all three study groups of healthy, undiagnosed hypertensive and known hypertensives; for healthy group systolic blood < 120 mmHg and diastolic < 80 mmHg taken twice on two different days; for patients with undiagnosed hypertension systolic blood pressure ≥ 130 mmHg or a diastolic blood pressure ≥ 80 mmHg measured twice on two different days without prior diagnosis of hypertension; and for hypertensive group patient taking anti-hypertensive medications with a known diagnosis of hypertension were included. Exclusion criteria for healthy controls and patients with undiagnosed hypertension were systemic disease, active treatment with immunosuppressant, chemotherapy, radiotherapy, pregnancy, or on any medication. Exclusion criteria for patients with known hypertension were any systemic disease other than hypertension, active treatment with immunosuppressant, chemotherapy, radiotherapy, or pregnancy. Individuals with elevated blood pressure (previously known as pre-hypertension) [20] and those with any systemic disease in healthy and undiagnosed hypertension groups and systemic disease other than hypertension in the hypertension group were excluded to reduce bias.

The variables included were age, sex (male/female), average systolic and diastolic blood pressure measurements, mean arterial pressure (MAP), perceived stress scores, and DMFT index. For socio-demographics and habits, highest educational attainment, habit of smoking, and alcohol consumption were also recorded.

### Blood pressure measurement

Blood pressure was recorded after 15 minutes of comfortable resting on two different days at the same time of the day where two recordings were taken 10 minutes apart (a total of four readings) for all patients by a dental out-patient department nurse. Blood pressure was recorded with Omron digital blood pressure monitor (M10-IT, Omron Healthcare, Kyoto, Japan). Based on average blood pressure readings, patient was either placed in healthy group if systolic blood pressure was < 120 mmHg and diastolic blood pressure was < 80 mmHg or in undiagnosed hypertensive group if systolic blood pressure of ≥ 130 mmHg or a diastolic blood pressure of ≥ 80 mmHg measured twice on two different days [4, 20]. Blood pressure readings were also measured of those who were known hypertensives following the same protocol. Based on systolic and diastolic readings, MAP was also calculated of all participants by adding systolic blood pressure and two times diastolic blood pressure divided by three.

### Assessment of perceived stress

PSS-10 questionnaire Urdu version was given to the patients [21, 22]. The PSS-10 is a valid and reliable stress assessment tool which determines an individual's perceived stress levels. It consists of 10 questions that inquire about the person's feelings concerning daily problems, personal events, and coping mechanisms in the previous month. The sum of the individual scores of the 10 questions results in an overall score out of 40, which categorizes the person's stress level as low (0–13), moderate (14–26), or high (27–40) [23].

## Assessment of oral health status

Assessment of oral health status was done by using DMFT index. Before the data collection, the three investigators recording DMFT were calibrated on 10 patients for intra-examiner and inter-examiner reliability. Inter-examiner and intra-examiner reliability were accessed by intra-class correlation coefficient which was above 0.8 for inter- and intra-examiner reliability. The DMFT score is a commonly used method for evaluating an individual's oral health status by dental examination using sterilized dental instruments such as mirrors, explorers, tweezers, excavators, gloves, and masks. The criteria for recording the DMFT score involve classifying decayed, missing, and filled teeth based on specific conditions, such as the presence of dental caries or restorations [24]. All 28 teeth were examined during this process. The overall DMFT score was recorded for all participants.

## Data analyses

Data was analyzed using SPSS version 27. Age, DMFT score, and PSS-10 score were expressed as mean and standard deviation (SD). Sex (male/female) and stress groups were computed as frequency and percentage. Comparison of age, PSS-10, DMFT, and MAP between healthy, patients with undiagnosed hypertension, and those with known hypertension were done by Kruskal Wallis H test because the continuous variables were not normally distributed as per Shapiro Wilk's test. For comparison of sex (male/female) and stress groups (mild, moderate, severe) with study groups (healthy, undiagnosed hypertensive, known hypertensive) was done by Chi-squared or Fisher's exact tests. Comparisons between PSS-10, DMFT, MAP, and age of healthy and patients with undiagnosed hypertension were done using Spearman's matrix correlation. A binary logistic regression analysis was conducted with presence of hypertension as dependent variable and sex, age, PSS-10, DMFT, smoking status, and alcohol use as covariates. For all these analyses, p value < 0.05 was considered statistically significant.

## Results

A total of 108 participants were included in this study, 36 in each group: healthy, undiagnosed hypertensive, and known hypertensive. The mean age of the participants was 48.84±11.84 years (age range 24–69 years). Males were 48 (44.4%) and females were 60 (55.6%) of the total participants. Characteristics of the study participants are given in Table 1. Mean DMFT score was 8.72±5.41 (range 1–28) and mean PSS-10 score (range 10–28) was 20.16±4.33 of the study participants. Among the 36 participants in the hypertensive group, 18 (50%) had uncontrolled hypertension (blood pressure >140/90 mmHg). Table 1 provides a comparison of sex and stress categories with the study groups. The association between sex and blood pressure groups

**Table 1. Characteristics of the study participants.**

| Characteristics | | n (%) | Healthy n (%) | Undiagnosed hypertensive n (%) | Known hypertensive n (%) |
|---|---|---|---|---|---|
| **Educational attainment** | Middle school (8 years) | 2 (1.9) | - | - | 2 (5.6) |
| | High school (10 years) | 32 (29.6) | 10 (27.8) | 16 (44.4) | 6 (16.7) |
| | Intermediate (12 years) | 46 (42.6) | 16 (44.4) | 10 (27.8) | 20 (55.6) |
| | Graduate (16 years) | 28 (25.9) | 10 (27.8) | 10 (27.8) | 8 (22.2) |
| **Smoking habit** | Yes | 8 (7.4) | 2 (5.6) | 6 (16.7) | - |
| | No | 100 (92.6) | 34 (94.4) | 30 (83.3) | 36 (100) |
| **Alcohol consumption** | Yes | 2 (1.9) | - | - | 2 (5.6) |
| | No | 106 (98.1) | 36 (100) | 36 (100) | 34 (94.4) |

**Table 2. Comparison of perceived stress and sex among the study participants.**

| Variable | | Total participants n (%) | Healthy n (%) | Undiagnosed hypertensive n (%) | Known hypertensive n (%) | p value |
|---|---|---|---|---|---|---|
| **Sex** | Male | 48 (44.4) | 16 (33.3) | 20 (41.7) | 12 (25) | 3.6 (2), 0.165 |
| | Female | 60 (55.6) | 20 (33.3) | 16 (26.7) | 24 (40) | |
| **Perceived stress** | Low | 15 (13.9) | 14 (93.3) | 1 (6.7) | - | <0.001* |
| | Moderate | 88 (81.5) | 22 (25) | 32 (36.4) | 34 (38.6) | |
| | High | 5 (4.6) | - | 3 (60) | 2 (30) | |

Chi-squared or Fisher's exact test used

*Significant at p < 0.05.

**Table 3. Comparison of age, DMFT and PSS-10 among the study participants.**

| Variable | Healthy Mean (SD) | Undiagnosed hypertensive Mean (SD) | Known hypertensive Mean (SD) | p value |
|---|---|---|---|---|
| **Age (in years)** | 41.89 (12.08) | 53.08 (10.84) | 51.56 (9.44) | <0.001* |
| **DMFT** | 5.89 (2.45) | 11.28 (6.12) | 9 (5.57) | <0.001* |
| **PSS-10** | 17.58 (4.97) | 21.08 (3.92) | 21.81 (2.63) | 0.003* |
| **MAP (mmHg)** | 89.18 (2.46) | 105.50 (5.43) | 104.79 (11.29) | <0.001* |

DMFT: Decayed, missing, filled teeth; PSS: Perceived stress scale; Kruskal Wallis H test

*Significant at p < 0.05.

was not significant. However, there were statistically significant differences in stress groups and study groups as given in Table 2.

Kruskal Wallis H tests were used to compare study groups with age, DMFT, PSS-10, and MAP as shown in Table 3. There were significant relationships between these variables and study groups. DMFT and perceived stress scores were higher in undiagnosed hypertensives and known hypertensives compared to healthy controls.

According to Spearman's matrix correlation test, moderate positive correlations were seen between MAP and DMFT, MAP and age, DMFT and perceived stress, and DMFT and age in healthy controls and suspected hypertensives, as given in Table 4. When correlation matrix test was applied on all study participants, moderate positive correlations were observed between MAP and perceived stress, MAP and age, MAP and DMFT, DMFT and perceived stress, and DMFT and age, as shown in Table 5.

**Table 4. Correlation between age, PSS-10, DMFT, and MAP between healthy and undiagnosed hypertensive participants.**

| Variable | DMFT (r, p value) | PSS-10 (r, p value) | Age (r, p value) |
|---|---|---|---|
| **PSS-10** | 0.45, <0.001* | | |
| **Age** | 0.33, 0.004* | 0.18, 0.127 | |
| **MAP** | 0.46, <0.001* | 0.19, 0.104 | 0.38, 0.001* |

DMFT: Decayed, missing, filled teeth; MAP: Mean arterial pressure; PSS: Perceived stress scale; Spearman's matrix correlation test used

*Correlation is significant at 0.01 level (2-tailed).

**Table 5. Correlation between age, PSS-10, DMFT, and MAP between healthy, undiagnosed hypertensive, and known hypertensive participants.**

| Variable | Age (r, p value) | MAP (r, p value) | DMFT (r, p value) |
|---|---|---|---|
| **MAP** | 0.40, <0.001* | | |
| **DMFT** | 0.30, 0.002* | 0.54, < 0.001* | |
| **PSS-10** | 0.15, 0.127 | 0.29, 0.002* | 0.38, < 0.001* |

DMFT: Decayed, missing, filled teeth; MAP: Mean arterial pressure; PSS: Perceived stress scale; Spearman's matrix correlation test used

*Correlation is significant at 0.01 level (2-tailed).

**Table 6. Logistic regression analysis of independent variables with outcome (presence of hypertension) in dental patients.**

| Presence of hypertension | Coefficient | p value | OR (95% CI) |
|---|---|---|---|
| Male | 0.40 | 0.475 | 1.49 (0.50–4.47) |
| Age | 0.09 | <0.001* | 1.09 (1.04–1.15) |
| DMFT | 0.18 | 0.028* | 1.19 (1.02–1.40) |
| PSS-10 | 0.22 | 0.002* | 1.24 (1.08–1.42) |
| Smoker | 2.04 | 0.160 | 7.72 (0.45–133.65) |
| $R^2$ | 49% | | |

DMFT: Decayed, Missing, Filled Teeth; PSS: Perceived Stress Scale; $R^2$: Nagelkerke R squared.

*Significant at p = 0.05 level.

A binary logistic regression analysis was conducted with dependent variable of presence of hypertension (undiagnosed and diagnosed) and the predictor variables added were age, sex, smoking habit, perceived stress scores, and DMFT. There was only one participant who used alcohol, hence; it was not added as predictor variable in the model. The model explained 49% of variance for presence of hypertension and correctly classified 78.7% of cases, with specificity of 61.1% and sensitivity of 87.5%. The result shows that age, perceived stress scores and DMFT were suitable predictors of presence of hypertension in our study participants as given in Table 6.

## Discussion

This study provides valuable insights into the connection between oral health, perceived stress, and elevated blood pressure, emphasizing the critical role dentists can play in identifying potential hypertensive patients. Majority of the participants in undiagnosed hypertensive group completed their high school education and those who were healthy and known hypertensives, have completed 12 years of higher secondary schooling. A large cohort study in Chinese adults showed that the incidences of newly diagnosed hypertension were 27%, 19.4%, and 16.1%, respectively, in the elementary school and below, middle school, and high school degree or above groups [25]. Others have also reported that hypertension is more prevalent in individuals who are less educated [26]. Comparison could not be drawn due to small sample size of this study. Majority of the participants were non-smokers and were not using alcohol. However, participants may have not disclosed their habits to the dentist because of fear of judgment [27].

Our findings demonstrate significant differences in DMFT among systemically healthy, undiagnosed hypertensive, and known hypertensive dental patients. Notably, patients with

undiagnosed hypertension and known hypertension exhibited higher DMFT scores compared to systemically healthy patients, suggesting an association between poor oral health and elevated blood pressure. This corroborates with previous studies [15, 19, 28, 29]. Higher mean DMFT scores in patients with pre-hypertension and hypertension compared to healthy controls were reported by others [15]. In this study, the undiagnosed hypertensive group had higher DMFT compared to known hypertensives which could be due to control of elevated blood pressure because of medications or lifestyle change. Routinely incorporating blood pressure screenings into dental appointments allows leveraging the frequency of such visits to identify and manage hypertension early, potentially mitigating its long-term oral health consequences [30, 31]. The observed correlation between MAP and DMFT scores aligns with previous research indicating that hypertension may play a role in deteriorating oral health [17, 32]. Elevated blood pressure can lead to alterations in blood flow and vascular functions, potentially exacerbating oral health problems such as dental caries and periodontal disease [33]. However, there are conflicting results by others where hypertension was not associated with poor oral health as assessed by dental treatment needs [18] or DMFT [15]. This could be because of different characteristics of the study population and smaller sample size. The positive correlation between MAP and age further underscores the impact of aging on both blood pressure and oral health, implying that older individuals are more susceptible to these conditions [18, 34].

In this study, majority of the participants reported moderate stress which corroborates with previous studies [13, 23]. It is interesting to note that almost all the participants with low stress levels were normotensives and only one of the healthy participants was in undiagnosed hypertensive group. Moreover, the mean perceived stress scores were also higher of patients with undiagnosed and known hypertension compared to healthy controls. The association between perceived stress and oral health, as demonstrated by the positive correlation between PSS-10 and DMFT scores, supports existing research on the adverse effects of stress on oral health [35]. The binary logistic regression showed that increase in age, PSS-10, and DMFT increase the risk of having hypertension. It is interesting to note that for every one-year increase in age and one unit increase in PSS-10 and DMFT, the odds of being hypertensive increased to 1.09, 1.24, and 1.19 times, respectively. Age, perceived stress, and DMFT have been reported to be associated with hypertension [2, 13, 18, 36]. Smoking was not a significant predictor of hypertension in the participants which corroborates with a longitudinal study conducted in Indonesia which also did not report an association between smoking and hypertension [37]. Others have also reported that heavy smoking increased the risk of hypertension but not light smoking [38]. There are conflicting results; some have reported that smoking is a risk factor of hypertension [2, 18, 39, 40]. The difference in results could be due to small sample size, incorrect disclosure by the participants, or not recording other variables such as type of tobacco, frequency of daily smoking, lifestyle, and general health status such as body mass index.

Prolonged stress can influence oral health behaviours, such as oral hygiene practices and dietary choices, and can also impact the immune system, making individuals more susceptible to oral diseases. Moreover, the elevated perceived stress levels observed in patients with undiagnosed and known hypertension corroborates with previous studies that stress is correlated with high blood pressure and is one of the risk factors of hypertension [3, 14]. The results also suggest that patients with known hypertension had poor management of their increased blood pressure, which would be because of not adhering to the treatment of hypertension [41]. High psychological stress has been associated with reduced salivary flow rate and xerostomia [42], and xerostomia is also associated with dental caries and poor oral health [43]. It highlights the need for comprehensive patient care that addresses both physical and psychological well-

being. Our findings suggest that stress management should be an integral component of oral health care, particularly for individuals with elevated blood pressure [44].

The noticeable differences in DMFT and PSS-10 scores across the three groups highlight the potential benefits of integrating routine blood pressure screenings into dental care settings. Identifying and referring patients with undiagnosed hypertension to medical professionals during dental visits can enable timely intervention, potentially lowering the risk of cardiovascular complications [45].

Dental clinics could implement protocols for routine blood pressure monitoring and stress assessments and establish referral pathways to medical professionals. Integrating dental and general health services can provide a more comprehensive care model, improving patient outcomes. This interdisciplinary approach may be particularly advantageous in settings with limited healthcare access, ensuring that patients receive holistic care and that conditions like hypertension are not overlooked [9, 46].

While this study offers important insights, it is not without limitations. The sample size was relatively small, and the study was carried out at a single centre, which may limit the generalizability of the findings. However, the number of participants in each group were increased from the calculated sample size of 16 to 36 to increase reliability. The use of purposive sampling rather than random sampling introduces potential bias, which could affect the study's validity. Random sampling would have enhanced the representativeness of the sample. Moreover, due to time constraints socio-demographic characteristics including monthly household income and physical characteristics such as body mass index and lifestyle were not recorded. It could also be possible that the participants were not aware of any systemic disease that they might be having. Future research with larger, more diverse populations, extended duration, and multi-centre studies with inclusion of other parameters of oral health such as periodontal assessment are needed to confirm these results and further explore the complex interactions between hypertension, oral health, and stress.

## Conclusion

Our research highlights the significant associations between elevated blood pressure, poor oral health, and increased stress. The findings suggest that dentists can play a crucial role in the early identification of hypertension by incorporating routine blood pressure screenings into dental visits. This proactive approach can lead to timely referrals to healthcare providers for further evaluation and management, thereby preventing complications associated with untreated hypertension, such as cardiovascular diseases, stroke, and kidney failure. Further research is needed to expand our understanding of these relationships and to develop effective strategies for improving health outcomes in hypertensive individuals.

## Supporting information

**S1 Dataset.**
(XLSX)

## Author Contributions

**Conceptualization:** Saira Atif.

**Data curation:** Uffaq Syed, Musarat Rafiq, Ateeb Fatima.

**Formal analysis:** Saira Atif, Sadia Rana.

**Funding acquisition:** Saira Atif.

**Investigation:** Uffaq Syed.

**Methodology:** Saira Atif, Uffaq Syed.

**Supervision:** Saira Atif.

**Writing – original draft:** Saira Atif, Uffaq Syed, Musarat Rafiq, Ateeb Fatima, Madiha Tariq.

**Writing – review & editing:** Saira Atif, Sadia Rana.

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
