## [Decision Letter · Decision Letter 0]

19 Aug 2024

PONE-D-24-30378Comparison of perceived stress and oral health status between healthy, suspected hypertensive, and known hypertensive dental patientsPLOS ONE

Dear Dr. Atif, 

Thank you for submitting your manuscript to PLOS ONE. After careful consideration, we feel that it has merit but does not fully meet PLOS ONE’s publication criteria as it currently stands. Therefore, we invite you to submit a revised version of the manuscript that addresses the points raised during the review process.

We look forward to receiving your revised manuscript.

Kind regards,

Hadi Ghasemi

Academic Editor

PLOS ONE

Journal Requirements:

2. We note that there is identifying data in the Supporting Information file <S1 dataset.xlsx>. Due to the inclusion of these potentially identifying data, we have removed this file from your file inventory. Prior to sharing human research participant data, authors should consult with an ethics committee to ensure data are shared in accordance with participant consent and all applicable local laws.

-Location data

Please remove or anonymize all personal information (Age), ensure that the data shared are in accordance with participant consent, and re-upload a fully anonymized data set. Please note that spreadsheet columns with personal information must be removed and not hidden as all hidden columns will appear in the published file.

Reviewers' comments:

Reviewer's Responses to Questions

**Comments to the Author**

1. Is the manuscript technically sound, and do the data support the conclusions?

Reviewer #1: Partly

Reviewer #2: Partly

Reviewer #3: Yes

Reviewer #4: Yes

2. Has the statistical analysis been performed appropriately and rigorously? 

Reviewer #1: Yes

Reviewer #2: I Don't Know

Reviewer #3: No

Reviewer #4: Yes

3. Have the authors made all data underlying the findings in their manuscript fully available?

Reviewer #1: Yes

Reviewer #2: No

Reviewer #3: Yes

Reviewer #4: Yes

4. Is the manuscript presented in an intelligible fashion and written in standard English?

Reviewer #1: Yes

Reviewer #2: Yes

Reviewer #3: Yes

Reviewer #4: Yes

5. Review Comments to the Author

Reviewer #1: Article is good in writing and structure. However the relevance of the topic to the disease described is not clearly referred to and it is only the health behavior which is addressed . Suggestion is to edit the title accordingly.

Reviewer #2: This study examines the association between oral health status, perceived stress, and hypertension among dental patients. Conducted at the CMH Lahore Medical College & Institute of Dentistry in Pakistan, the research involved 108 participants, divided into three groups: healthy, suspected hypertensive, and known hypertensive patients. Using the Decayed, Missing, and Filled Teeth (DMFT) index and the Perceived Stress Scale (PSS-10), the study found significant correlations between elevated blood pressure, higher perceived stress, and deteriorated oral health. The study emphasizes the potential role of dentists in identifying undiagnosed hypertension through routine screenings during dental visits.

Critique:

1. Sample Size and Generalizability: The study's relatively small sample size and its focus on a single institution limit the generalizability of the findings. A larger, multi-center study would provide more robust and widely applicable results.

2. Sampling Method: The use of purposive sampling rather than random sampling introduces potential bias, which could affect the study’s validity. Random sampling would have enhanced the representativeness of the sample.

3. Lack of Control for Confounding Variables: The study does not account for potential confounding factors such as socioeconomic status, lifestyle factors (like diet and exercise), and other health conditions that could influence both stress and oral health. This omission weakens the conclusions drawn about the direct relationship between hypertension, stress, and oral health.

4. Measurement Tools: While the DMFT index and PSS-10 are established tools, the study could benefit from a broader assessment of oral health and stress. Including measures like periodontal health or using more detailed psychological assessments might offer a more comprehensive understanding of the relationships explored.

5. Limited Discussion on Mechanisms: The discussion touches on possible links between hypertension and oral health but lacks a deeper exploration of the biological or behavioral mechanisms that might explain these relationships. Providing more context from existing literature would strengthen the study’s theoretical foundation.

6. Short Study Duration: The study was conducted over just two months, which may not adequately capture the variations in stress and oral health that can occur over longer periods. Longitudinal studies would be better suited to observe the long-term effects of stress and hypertension on oral health.

7. Absence of Longitudinal Data: The cross-sectional design limits the ability to draw causal inferences. Future research could benefit from a longitudinal approach to better establish the temporal relationship between hypertension, stress, and oral health deterioration.

Overall, while the study offers valuable insights into the potential role of dentists in early hypertension detection, addressing these limitations in future research would enhance the reliability and impact of the findings.

Reviewer #3: The paper evaluated the perceived stress (by PSS-10 questionnaire) and oral health status (by using DMFT index) between patients with hypertension. The respected Authors categorized patients into healthy, suspected hypertensive, and known hypertensive dental patients. Overall, the article is well-written and has used validated questionnaires for obtaining the needed metrics. Since the association was examined using Chi-square test and correlation, the yielded links could be a result some variables that have not been adjusted for.

Although important studies have been conducted on the association of perceived stress and oral hygiene care separately, the novelty of the present study is to conduct the present study on patients into healthy, suspected hypertensive, and known hypertensive patients.

Additionally, there are some issues that I would like address.

1. The respected authors have used the term “suspected hypertension” which is not quit right. Based on the explained method, this includes patients with two times SBPs >130 or DBP>90. This would translate to undiagnosed HTN rather than suspects. This change would also mandate some revisions for the rationales stated for the rate of oral health status in this population in the discussion section.

2. I recommend using “patients with hypertension” instead of “hypertensive patients” across the manuscript as it has negative burden.

3. Please provide the exact purposive sampling method in line 85 as it can result in selection bias.

4. Lines 114- 117 are repeated and can be removed.

5. The reference for validation of perceived stress, is based on age 19-30 years. As older population exist in the current study, I suggest elaborating it as A limitation of the study.

6. In method section, were the continuous variables normally distributed? I’d suggest providing the appropriate statistical metrics for them.

7. Table 1 and 2 have the duplicate data due to the gender of the population in the study.

8. In table 1 and result section, I’d highly recommend to add three extra columns reporting the characteristics based on the hypertension groups and incorporating them accordingly to the discussion section, as well.

9. Caution should be used when definitive association or equivalent terms are used in the result or discussion section, as the probable role of other possible co variates was not accounted for in this article. So, if possible, conducting a regression, with adjusting for confounding variables could significantly improve the conclusions of the study.

10. In line 197, please provide more detail to compare your finding with other studies.

Reviewer #4: The manuscript is an important one and is well written.

However,

1. Reference is needed for line 44.

2. One or more references should be included in Page 3 line 60, page 7 line 214 since studies were mentioned.

6. PLOS authors have the option to publish the peer review history of their article (what does this mean?). If published, this will include your full peer review and any attached files.

Reviewer #1: **Yes: **Ghassem Ansari

Reviewer #2: No

Reviewer #3: **Yes: **Fatemeh Farshad

Reviewer #4: No

---

## [Author Response · Author response to Decision Letter 0]

10 Sep 2024

We have incorporated the suggestions into the revised manuscript. Thank you for your valuable comments and suggestions. A detailed response and the changes incorporated are given below:

Suggestions by Academic Editor: 

Thank you for highlighting a very important aspect related to anonymity of the data. We have removed age column from the data. 

Comments 1-4: 

We thank the reviewers for their valuable time and for constructive feedback which has undoubtedly improved the revised manuscript. 

Comment 5:

For Reviewer 1: We thank you for your review. For minor grammatical errors, these have been corrected on page 6, 7, 15, 17 and 18. Kindly let me know if there is anything specific that I might have missed. 

For Reviewers 2,3,4: We are thankful for your valuable suggestions and comments working on which have undoubtedly improved the manuscript. We would highly appreciate if you could go through the revised manuscript and guide further if more corrections are needed. The following corrections are done:

Reviewer’s suggestion 

Reviewer 1: 

Article is good in writing and structure. However, the relevance of the topic to the disease described is not clearly referred to and it is only the health behavior which is addressed. Suggestion is to edit the title accordingly A: The title of the manuscript has been revised. Please let us know if the revised title is appropriate. Page 1 and 2

Reviewer 2: 

1. Small sample size and its focus on a single institution limit the generalizability of the findings. A larger, multi-center study would provide more robust and widely applicable results. 

A: This has been added in the last paragraph of discussion as the limitation of this study. line 285-297

2. The use of purposive sampling rather than random sampling introduces potential bias, which could affect the study’s validity. Random sampling would have enhanced the representativeness of the sample. 

A: It has been incorporated as study limitation line 288-291

3. Lack of control for confounding variables. The study does not account for potential confounding factors such as socioeconomic status, lifestyle factors (like diet and exercise), and other health conditions that could influence both stress and oral health. This omission weakens the conclusions drawn about the direct relationship between hypertension, stress, and oral health. 

A: We have added table 6 for binary logistic regression analysis which accounts for the adjustment of possible confounders. Other limitations were mentioned in the last paragraph of discussion. Thank you for your valuable suggestions. line 156-158

4. Measurement Tools: While the DMFT index and PSS-10 are established tools, the study could benefit from a broader assessment of oral health and stress. Including measures like periodontal health or using more detailed psychological assessments might offer a more comprehensive understanding of the relationships explored. 

A: It went beyond our study scope. However, we have included the suggestions in limitations and future recommendations line 294-297

5. Limited Discussion on Mechanisms: The discussion touches on possible links between hypertension and oral health but lacks a deeper exploration of the biological or behavioral mechanisms that might explain these relationships. Providing more context from existing literature would strengthen the study’s theoretical foundation. 

A: We have added more details in discussion session about possible link between hypertension, stress and oral health. line 274-276

6. Short Study Duration: The study was conducted over just two months, which may not adequately capture the variations in stress and oral health that can occur over longer periods. Longitudinal studies would be better suited to observe the long-term effects of stress and hypertension on oral health. 

A: We have included this as limitation. line 294-297

7. Absence of Longitudinal Data: The cross-sectional design limits the ability to draw causal inferences. Future research could benefit from a longitudinal approach to better establish the temporal relationship between hypertension, stress, and oral health deterioration. 

A: We have included this as limitation and future recommendations line 294-297

Reviewer 3 

1. The respected authors have used the term “suspected hypertension” which is not quit right. Based on the explained method, this includes patients with two times SBPs >130 or DBP>90. This would translate to undiagnosed HTN rather than suspects. This change would also mandate some revisions for the rationales stated for the rate of oral health status in this population in the discussion section. 

A: We used suspected hypertension as we being dentists were not able to diagnose hypertension on our own and were in a position to refer such patients for proper diagnosis form a medical specialist. We have rephrased suspected hypertension with undiagnosed hypertension across the manuscript. All

2. I recommend using “patients with hypertension” instead of “hypertensive patients” across the manuscript as it has negative burden. 

A: Thank you for your valuable input. It has been corrected across the manuscript. All

3. Please provide the exact purposive sampling method in line 85 as it can result in selection bias. 

A: It is added. Line 91-94

4. Lines 114-117 are repeated and can be removed. 

A: It has been rephrased and repetition removed. Line 119

5. The reference for validation of perceived stress, is based on age 19-30 years. As older population exist in the current study, I suggest elaborating it as A limitation of the study 

A: We have added another reference for perceived stress in comparative age of 18-69 years. Thank you for highlighting this. line 129

6. In method section, were the continuous variables normally distributed? I’d suggest providing the appropriate statistical metrics for them. 

A: It has been added that continuous variables were not normally distributed as per Shapiro Wilk’s test. Hence, non-parametric Kruskal Wallis test was used for comparison. line 149-152

7. Table 1 and 2 have the duplicate data due to the gender of the population in the study. 

A: Sex from table 1 has been removed. Table 1 page 7

8. In table 1 and result section, I’d highly recommend to add three extra columns reporting the characteristics based on the hypertension groups and incorporating them accordingly to the discussion section, as well. 

A: Table has been modified and relevant discussion added. Table 1 page 7, line 216-225

9. Caution should be used when definitive association or equivalent terms are used in the result or discussion section, as the probable role of other possible co variates was not accounted for in this article. So, if possible, conducting a regression, with adjusting for confounding variables could significantly improve the conclusions of the study. 

A: We have conducted binary regression analysis and have included it in results and discussion. line 198-205, Table 6, 255-266

10. In line 197, please provide more detail to compare your finding with other studies. 

A: We have included another reference. line 225, 258-259

Reviewer 4 

1. Reference is needed for line 44. 

A: Thank you for highlighting very important point. Reference is added. line 47

2. One or more references should be included in Page 3 line 60, page 7 line 214 since studies were mentioned. 

A: References are added. line 64, 243

---

## [Decision Letter · Decision Letter 1]

24 Sep 2024

Comparison of perceived stress and oral health status using Perceived Stress Scale and DMFT index between healthy, undiagnosed hypertensive, and known hypertensive dental patients

PONE-D-24-30378R1

Dear Dr. Saira Atif

We’re pleased to inform you that your manuscript has been judged scientifically suitable for publication and will be formally accepted for publication once it meets all outstanding technical requirements.

Kind regards,

Hadi Ghasemi

Academic Editor

PLOS ONE

Additional Editor Comments (optional):

Reviewers' comments:

Reviewer's Responses to Questions

**Comments to the Author**

1. If the authors have adequately addressed your comments raised in a previous round of review and you feel that this manuscript is now acceptable for publication, you may indicate that here to bypass the “Comments to the Author” section, enter your conflict of interest statement in the “Confidential to Editor” section, and submit your "Accept" recommendation.

Reviewer #3: All comments have been addressed

Reviewer #4: All comments have been addressed

2. Is the manuscript technically sound, and do the data support the conclusions?

Reviewer #3: Yes

Reviewer #4: Yes

3. Has the statistical analysis been performed appropriately and rigorously? 

Reviewer #3: Yes

Reviewer #4: Yes

4. Have the authors made all data underlying the findings in their manuscript fully available?

Reviewer #3: Yes

Reviewer #4: Yes

5. Is the manuscript presented in an intelligible fashion and written in standard English?

Reviewer #3: Yes

Reviewer #4: Yes

6. Review Comments to the Author

Reviewer #3: I read the paper entitled “Comparison of perceived stress and oral health status using Perceived Stress Scale and DMFT index between healthy, undiagnosed hypertensive, and known hypertensive dental patients” with great interest and congratulate the honorable authors for their excellent work. The article is well-written and has a meticulously answered all previous comments. Just I have to mentioned that Table 6 does not have any title. Please consider that as well.

Reviewer #4: All comments have been addressed. The manuscript is technically sound and data analysis good. The manuscript is written in standard English.

7. PLOS authors have the option to publish the peer review history of their article (what does this mean?). If published, this will include your full peer review and any attached files.

Reviewer #3: **Yes: **Fatemeh Farshad

Reviewer #4: No

---

## [Editor Report · Acceptance letter]

1 Oct 2024

PONE-D-24-30378R1 

PLOS ONE

Dear Dr. Atif, 

I'm pleased to inform you that your manuscript has been deemed suitable for publication in PLOS ONE. Congratulations! Your manuscript is now being handed over to our production team.

Kind regards, 

on behalf of

Dr. Hadi Ghasemi 

Academic Editor

PLOS ONE